# Specificity Proteins (Sp) and Cancer

**DOI:** 10.3390/ijms24065164

**Published:** 2023-03-08

**Authors:** Stephen Safe

**Affiliations:** Department of Veterinary Physiology and Pharmacology, Texas A&M University, College Station, TX 77843, USA; ssafe@cvm.tamu.edu

**Keywords:** Sp1, Sp3, Sp4, non-oncogene addiction, prognostic, pro-oncogenic

## Abstract

The specificity protein (Sp) transcription factors (TFs) Sp1, Sp2, Sp3 and Sp4 exhibit structural and functional similarities in cancer cells and extensive studies of Sp1 show that it is a negative prognostic factor for patients with multiple tumor types. In this review, the role of Sp1, Sp3 and Sp4 in the development of cancer and their regulation of pro-oncogenic factors and pathways is reviewed. In addition, interactions with non-coding RNAs and the development of agents that target Sp transcription factors are also discussed. Studies on normal cell transformation into cancer cell lines show that this transformation process is accompanied by increased levels of Sp1 in most cell models, and in the transformation of muscle cells into rhabdomyosarcoma, both Sp1 and Sp3, but not Sp4, are increased. The pro-oncogenic functions of Sp1, Sp3 and Sp4 in cancer cell lines were studied in knockdown studies where silencing of each individual Sp TF decreased cancer growth, invasion and induced apoptosis. Silencing of an individual Sp TF was not compensated for by the other two and it was concluded that Sp1, Sp3 and Sp4 are examples of non-oncogene addicted genes. This conclusion was strengthened by the results of Sp TF interactions with non-coding microRNAs and long non-coding RNAs where Sp1 contributed to pro-oncogenic functions of Sp/non-coding RNAs. There are now many examples of anticancer agents and pharmaceuticals that induce downregulation/degradation of Sp1, Sp3 and Sp4, yet clinical applications of drugs specifically targeting Sp TFs are not being used. The application of agents targeting Sp TFs in combination therapies should be considered for their potential to enhance treatment efficacy and decrease toxic side effects.

## 1. Background

Specificity protein 1 (Sp1) was among the first transcription factors (TFs) identified and is a member of the Sp/Kruppel-like factor (Sp/KLF) family. Members of this family exhibit variable structural domains and functions but all contain conserved zinc fingers in their DNA binding domains that bind GC-rich (Sps) and CACC (KLFs) boxes [1,2,3,4,5,6,7]. Not surprisingly, within the Sp and KLF sub-families there can be some overlap and competition for the same cis-elements, although for many Sp-regulated genes, differences in cell context and levels of expression dictate which Sp transcription factor is active. Among the 9 Sp genes, Sp1-Sp4 are most similar in terms of both structure and function (Figure 1), and they are the prime focus of this review. It should also be pointed out that among Sp1-Sp4, most research has focused on Sp1 and to a lesser extent Sp3 and it is possible that for some genes and pathways, the potential contributions of Sp2 and Sp4 have been understudied. There has been extensive research on the mechanisms of Sp-regulated gene expression, which frequently is observed in genes that lack a TATA box. Many Sp-regulated genes bind and activate gene expression through one or more GC-rich sequences proximal to the start sites where there are ordered assemblies of nuclear cofactors to form a transcriptionally active complex that includes DNA-bound Sp1, Sp3 or Sp4. The composition of transcription complexes includes polymerase II, transcription factor IID (TFIID), TATA box binding protein (TBP) and associated factors (TAFs) and members of the cofactor required for Sp1 activation/mediator (CRISP/MED) complexes [8]. The overall complex is highly variable and both gene- and cell-context-dependent. Moreover, there is also evidence that Sp TFs bind imperfect/variable GC-rich sequences and also interact with distal enhancer sequences, as described for the Topoisomerase IIa promoter [9]. In this review, there is a focus on the interactions of Sp TFs with non-coding RNAs and their functions; however, it should also be noted that Sp1 physically interacts with over 55 other proteins [2]. Sp1 function is also influenced by post-transcriptional modifications that include phosphorylation, acetylation, glycosylation and cleavage, and these changes can enhance or inhibit protein stability. Unfortunately, data for Sp3-Sp4 in terms of transcriptional function, post-transcriptional modifications and interactions with other factors have not been extensively investigated.

Several excellent reviews on Sp transcription factors and their role on genes and pathways associated with cancer and non-cancer endpoints have been published [1,2,3,4,5,6,7] and this article primarily focuses on Sp TFs and cancer findings from studies published within the last 5 years, more recent studies and their significance. It will become apparent that while Sp TFs are not oncogenes, their designation as non-oncogene addiction genes is highly appropriate [7].

## 2. Sp TFs as Cancer Prognostic Factors

Extensive analysis of tumor and non-tumor tissues has identified many prognostic factors that can be used to predict patient outcomes. Moreover, in some cases, the results dictate the application of specific treatment regimens, and this is particularly true of early-stage breast cancer where expression of estrogen receptor α (ERα, ESR1) in mammary tumors usually results in treatment with endocrine therapies [10]. Table 1 illustrates the important role of Sp1 as a negative prognostic factor for multiple cancers where Sp1 is generally more highly expressed in tumors compared to normal tissue and overexpression is correlated with decreased disease-free patient survival or another negative outcome. With the exception of highly variable results for lung cancer, most tumors overexpress Sp1 (or Sp3) compared to non-tumor tissue and poorer outcomes are observed in patients with tumors overexpressing this TF. In liver cancer, both Sp1 and Sp2 are negative prognostic factors for survival [11,12,13]. In many cases, manuscripts reporting the role of Sp1 as a diagnostic factor are accompanied by laboratory studies showing the pro-oncogenic functional activities of Sp1.

Meta-analysis of multiple studies has also been used to probe the role of Sp1 in gastric cancer, and higher Sp1 expression is correlated with increased depth of invasion and lymph node metastasis, increased TNM staging and Lauren’s classification [41]. A similar meta-analysis approach was used to examine multiple tumor types [42] and similar associations were observed as reported for gastric cancer.

## 3. Role of Sp in Cell Transformation

Sp1 is clearly a negative prognostic factor for multiple cancers, and this is accompanied by increased expression of Sp transcription factors in tumors compared to non-tumor tissues. These observations suggest that the process that drives the transformation of a normal cell to a tumor cell may also involve Sp transcription factors. This was investigated in a classical study that examined the effects of carcinogen or oncogene-induced transformation of human fibroblasts into fibrosarcoma cells in which the fibrosarcoma, but not the fibroblasts, had the ability to form tumors in athymic nude mice [43,44]. This dramatic change in the phenotype of fibrosarcoma cells compared to the fibroblasts was accompanied by an 8- to 18-fold increased expression of Sp1 protein, which is enhanced during fibroblast cell transformation. Moreover, it was also demonstrated that knockdown of Sp1 in the fibrosarcomas resulted in cells that did not form tumors in athymic nude mice. Other studies show that EGF-induced transformation of bladder epithelial cells and Kras induced transformation of MCFI0A cells also involved Sp1 or an Sp1-regulated gene [45,46]. CYP1B1 also enhanced the proliferation, migration and invasion of MCFI0A and MCF7 cells and this was also accompanied by increased expression of Sp1 and Sp1 regulated genes and silencing or inhibition of Sp1 inhibited CYP1B1-mediated transformation [47].

Arsenic is a carcinogen and considered to be a public health hazard. Exposures of human bronchial epithelial Beas-2B cells to arsenic over a period of several months lead to cell transformation and this was due, in part, to induction DNA methyltransferase 1 (DNMT1) [48]. However, further examination found that arsenic induced Sp1, which in part enhanced DNMT1 expression and loss of miR-199a-5p, which was critical for arsenic-induced transformation. The proposed mechanism involves arsenic-induced Sp1, which in turn activates DNMT1 and suppresses miR-199a-5p. These results demonstrate a role for Sp1 in arsenic-induced transformation of Beas-2B cells; however, the direct effect of Sp1-mediated suppression of miR-199a-5p is unexpected and needs further investigation. Rhabdomyosarcomas (RMS) express high levels of Sp1 compared to non-transformed muscle tissue and RMS cell lines express high levels of Sp1, Sp3 and Sp4. Transformation of human smooth muscles with telomerase, the PAX3-FOXO1 oncogene and NMyc transforms these muscle cell lines; however, expression of only one or two of these factors is not sufficient for transformation [49]. Interestingly, transfection of one or two of these genes dramatically induces expression of Sp1 and Sp3 but not Sp4. This suggests that the process of cell transformation is accompanied by early induction of Sp1 and Sp3 prior to conversion of the muscle cell into a cancer cell [50].

The role of Sp TFs in the process of transformation has also been investigated in cancer stem cells, where they directly regulate genes associated with “stemness” or cooperate with other genes and non-coding RNAs to enhance stemness. At present, there is strong evidence for the role of Sp1 in inducing stemness, and the cooperating factors vary with tumor type. Stemness in breast cancer is maintained by the long non-coding RNA408 (Lnc408)—dependent recruitment of Sp3 to CBY1 gene promoters to inhibit expression of CBY1, which indirectly enhances levels of nuclear β-catenin and β-catenin regulated cancer stem cell-related genes [51]. In gastric cancer, Sp1 regulates expression of leucine-rich repeat-containing receptor 5 (LGR5), a key stem cell factor [52], and in hepatocellular carcinoma, Sp1 induced LncRNA DPPA2 upstream binding RNA (DUBR) [53]. DUBR not only promotes stemness, but also oxaliplatin resistance through an Sp1/DUBR/E2F1-CIP2A axis. The cancer stem-cell-related protein BMI1 is overexpressed in lung cancer and is important for maintaining this phenotype and resistance to pemetrexed [54]. BMI1 also regulates Sp1 expression and knockdown of Sp1 or treatment mithramycin reverses many of the effects of BMI1, including chug resistance. The pro-oncogenic LncRNA HOTAIR interacts with and upregulates Sp1, which induces DNMI1, and transcriptional repression of miR-199a-5p and targeting downregulation of Sp1 or DNMI1 was found to decrease stemness and progression of cutaneous squamous cell carcinoma [55]. In papillary thyroid carcinoma, the LncRNA DOCK9-AS2 interacts with and induces Sp1, which in turn induces β-catenin, which is further induced by DOCK9-AS2 interacting with miR-1972, resulting in increased β-catenin and Wnt signaling [56]. Sp1 is overexpressed in glioblastoma cells [18,19,20] and plays a role in maintaining stemness and drug resistance in this tumor type. It was also reported that ANGPTL4 and Sp4 were overexpressed in GBM and predicted poor patient prognosis [57]. Sp4 also regulates ANGPTL4 and downstream EGFR/AKT/4E-BP1, which is associated with temozolomide resistance and expression of cancer stem cell markers. Drug resistance and stemness in GBM were also associated with Sp1 in another study [58] and in glioma, HDAC/Sp1 regulation of BMI1 enhanced stemness [59]; this exhibited some overlap with lung cancer cells and BMI1 [54].

## 4. Sp TFs and Regulation of Protein-Encoding Genes in Cancer Cells

In 1983–1984, Tjian and coworkers initially identified Sp1 as a factor that stimulated SV40 early promoter transcription by 40-fold and bound to GC-rich elements in target gene promoters [60,61]. This same group also identified Sp2 as another TF that bound SV40 [60], and approximately a decade later, Sp3 and Sp4 were also characterized [62,63,64,65,66] as a structurally related sub-class of the Sp/KLF family. Subsequent research has demonstrated that Sp1-Sp4 TFs directly regulate or co-regulate thousands of protein-encoding genes associated with cell proliferation, survival, migration and invasion [7]. A detailed study of the role of Sp1, Sp3 and Sp4 in cancer was investigated in multiple cancer cell lines by individual knockdown of the three genes and their combination coupled with analysis of the resulting functional and genomic effects and their overlap [66]. Knockdown of Sp1 (siSp1), Sp3 (siSp3) and Sp4 (siSp4) and their combination (siSp1, 3, 4) decreased growth, increased Annexin V staining (apoptosis) and decreased invasion in A549 lung, MiaPaca2 (pancreatic), SW480 (colon), 786-0 (kidney), SKBR3 (breast), MDA-MB231 (breast), Panc1 (pancreatic) and L3.6 pL (pancreatic) cancer cells. Knockdown efficiencies were high and cell context-dependent differences in functional response potencies were < three-fold for most responses. For most responses, cells deficient in Sp1, Sp3 and Sp4 (triple knockdown) exhibited the highest effect on growth inhibition, induction of Annexin V staining and inhibition of invasion; however, the magnitude of the differences between single and triple knockdown was relatively modest. These results indicate that Sp1, Sp3 and Sp4 individually regulate proliferation, survival and invasion of cancer cells and the loss of one of these TFs is not compensated or rescued by the other two. One possible explanation is that Sp1, Sp3 and Sp4 cooperatively regulate many of the same pro-oncogenic genes and loss of a single TF compromises any possible rescue by the other two.

The highly invasive Panc1 pancreatic cancer cell line was used as a model to investigate the differential expression of genes after knockdown of Sp1, Sp3 and Sp4. Figure 2 illustrates the number of DEGs after knockdown of Sp1, Sp3 and Sp4, including 3532, 4826 and 4232 genes, respectively. Further analysis shows that the common DEGs after knockdown of Sp1/Sp3, Sp1/Sp4 and Sp3/Sp4 were 1113, 1140 and 2753, respectively, indicating that pairs of the three Sp TFs regulated a relatively high percentage of genes in common. This was particularly true for Sp3/Sp4, in which 2753 genes were commonly regulated by both transcription factors, which includes 57 and 64% of all Sp3 and Sp4 regulated genes, respectively. This would suggest that particularly for Sp3 and Sp4 and also the other pairs (Sp1/Sp3, Sp1/Sp4), there may be significant cooperative regulation of genes that requires more than one Sp TF. As demonstrated in Figure 2 and Figure 3, Sp1, Sp3 and Sp4 regulate expression of several thousand genes, with many of them associated with cancer proliferation, survival, and migration/invasion. Moreover, the three transcription factors also regulate genes in common and also genes that are Sp- specific and vary with cell context. Sp (Sp1, Sp3 and Sp4) regulated genes include epidermal growth factor receptor 1 (EGFR), other tyrosine kinases, cMyc, bcl2, survivin, vascular endothelial growth factor receptors (VEGFR1 and VEGFR2), matrix metalloproteinases and many other genes.

Since Sp TF regulate genes associated with cell proliferation, survival, and invasion, we used ingenuity pathway analysis (IPA) to analyze DEGs for each pathway after knockdown of individual Sps and their combination. The relative expressions of DEGs were determined and the results are illustrated in Figure 3. The patterns of DEGs associated with Panc1 cell proliferation, survival, and invasion after knockdown of Sp1, Sp3 and Sp4 were similar; however, the number of genes involved followed the order of proliferation ≥cell death > invasion. In addition, the pattern of the number of DEGs commonly expressed by Sp1/Sp3, Sp1/Sp4 and Sp3/Sp4 associated with cell proliferation, survival and invasion was higher than that observed for the total genes. The percentage of common genes/total genes was the highest for Sp3/Sp4, where the percentages were 67%, 68% and 74% (Sp3), and 66%, 67% and 72% (Sp4) for cell proliferation, survival, and invasion respectively. Casual IPA analysis also confirmed by their z scores (>2.0 or <−2.0) that the DEGs in each group were strongly associated with the functional responses.

There is evidence from the large number of publications that not only do Sp1, Sp3 and Sp4 regulate pro-oncogenic pathways and genes, but there are also reports that Sp2 performs similar functions [13,67,68]. For example, Sp2 knockdown in hepatocellular carcinoma cells decreases cell migration, proliferation and survival of hepatocellular carcinoma cells and this is due, in part, to decreasing the expression of the TRIB3 gene [13]. Additionally, Sp2-dependent suppression carcinoembryonic antigen-related cell adhesion molecule 1 (CEACAM1) [67] and overexpression of Sp2 increase susceptibility to wound- and carcinogen-induced tumorigenesis [68]. Thus, Sp1-Sp4 regulation of protein-encoding genes plays an important role in cell transformation and tumorigenesis.

## 5. Sp TFs-MicroRNA (miRNA) Interactions in Cancer Cells

Although noncoding RNAs have been described long before the sequence of the human genome was published, it became evident from the sequencing data that only a small faction (1–2%) of the human genome encodes for proteins [69]. Subsequent studies have identified many different types of non-coding RNAs (ncRNAs), including housekeeping and regulatory ncRNAs, which have been linked to many functions, some of which include interactions with Sp TFs [69,70,71,72,73]. Mature miRNAs have a length ≤20 nucleotides and are processed from pri-miRNA; one of their major functions involves interactions of the seed sequences of these miRNAs with complementary 6-8 base pair sites in the 3′-region of target genes to inhibit transcription [70]. There is a sub-set of miRNAs that directly inhibit Sp1 expression and the resulting inverse expression of these miRNAs with Sp1 is sometimes also associated with their use as a positive prognostic value for cancer patients. MiRNAs that repress expression of Sp1, Sp3 and Sp4 are illustrated in Table 2, and it is clear that several miRNAs are key regulators of Sp expression in multiple tumor types and Sp1 is preferentially targeted in cancer cells. It is also evident that multiple miRNAs target Sp in the same tumor cell type. For example, miRNA-375, miRNA-375-3p, miRNA-1224-5p, miRNA-382, and miRNA-149 target Sp1 and decrease expression of Sp1 in colorectal cancer and eight miRNAs decrease Sp1 expression in gastric cancer. Some of the miRNAs in Table 2 and others are also regulated by Sp TF in cancer cells. For example, Sp1 induces expression of multiple miRNAs in lung cancer cells (miRNA-3194-5p, miRNA-218-5p, miRNA-193-5p, miRNA-182-5p and miRNA-135-5p), [74] miRNA-200 in breast cancer cells [75], and miRNA-365 in Hela cells [76]. In contrast, Sp1 decreases miR-335 expression in ovarian cancer cells, and this is one of the rare reported examples of Sp1 as a transcriptional receptor [77].

## 6. Sp TFs-LncRNA Interactions in Cancer Cells

Long non-coding RNAs (lncRNAs) are another class of ncRNAs that are > 200 nucleotides long, and it is estimated that the human genome encodes more than 28,000 lncRNAs. LncRNAs have multiple functions, including both tumor suppressor and tumor promoter-like activities [106,107,108,109]. These activities are the result of their diverse mechanisms of action that act via signaling, decoys, guides and scaffolds [110]. Sp1 plays a varied role in regulating LncRNA since Sp1 and various LncRNAs regulate each other individually or reciprocally and also cooperate with other gene products and miRNAs in cancer cells. Since Sp1 is a negative prognostic factor for many tumors, it is not surprising that Sp1 regulates expression of several LncRNA, many of which are also pro-oncogenic. Table 3 summarizes a number of lncRNAs that are directly regulated by Sp1 and some of these ncRNAs are also regulated by Sp3 and Sp4 [86,111,112,113,114,115,116,117,118,119,120,121,122,123,124,125,126,127,128,129,130,131,132,133,134,135,136,137,138,139,140,141,142,143]. Sp TFs also interact with lncRNA/miRNA where there is not a direct modulation of lncRNA/Sp expression [144,145,146,147,148]. In addition, there is also evidence that lncRNA LOC90024 promotes an RNA splicing step that results in formation of a long pro-oncogenic form of Sp4 [148,149]. Examples of mechanisms involving Sp TFs and lncRNAs include the following; direct transcriptional activation of lncRNAs by Sp1 (Figure 4A); sponging of miR-375 by RP11-626G11-3 to enhance Sp1 levels (Figure 4B); formation of an Sp1/XLOC013218 complex on the PIK3R2 promoter to activate gene expression (Figure 4C); and formation of an HDAC3/Sp1/EZH2 complex on the MEG3 promoter to inhibit gene expression (Figure 4D). The physical and functional interactions of Sp1, Sp3 and Sp4 with non-coding RNAs have primarily been observed for Sp1, as indicated from Table 2 and Table 3. However, it is apparent from the current available data that Sp interactions with ncRNAs are highly variable and cell-context-dependent. The emergence of dominant Sp-miRNA and Sp-lncRNA complexes that modulate critical pathways in cancer will be dependent on the results of future research. Thus, many functional effects of lncRNAs are Sp1 dependent and these are often in association with other genes involved in the complex. With few exceptions, lncRNA/miRNA pathways that lead to higher expression of Sp1, Sp3 and Sp4 result in downstream activation of pro-oncogenic genes/pathways, indicating that drugs targeting ncRNAs or Sp TFs should be highly effective anti-cancer agents.

## 7. Sp Transcription Factors as Drug Targets

Sp transcription factors are prognostic indicators for multiple cancers (Table 1) and interact with both miRNAs and lncRNAs (Table 2 and Table 3) to facilitate cancer cell proliferation, survival, migration and invasion. These pro-oncogenic activities correlate with the results of knockdown studies that are consistent with their designation as non-oncogene addiction genes [66]. Despite these facts, anticancer agents that specifically target SpTFs are not being developed currently for clinical applications, even though several small molecules that are used for cancer and other chemotherapies also downregulate/degrade Sp1, Sp3 and Sp4. These include HDAC inhibitors, metformin, bardoxolone methyl, bortezomib and some non-steroidal anti-inflammatory drugs (NSAIDs). Two review articles from this laboratory have previously outlined compounds that downregulate or induce degradation of Sp TFs [4,150], and these include drug-induced ROS, proteasome-dependent degradation, cannabinoid receptor (CBR) induced responses, zinc depletion and kinase/phosphatase pathways. Studies in this laboratory have investigated drugs that activate most of these pathways [4,150] and result in coordinated downregulation of Sp1, Sp3 and Sp4. Most other studies have focused on drug-induced downregulation of only Sp1, and it can be assumed that in many cases, downregulation of Sp1 is accompanied by parallel decreases in Sp3 and Sp4. Multiple classes of compounds decrease expression of Sp TFs in cancer cells, and these include structurally diverse ROS inducers, non-steroidal anti-inflammatory drugs (NSAIDs), cannabinoids and other drugs including retinoids, α—tocopherol thiazolidinediones, bortezomib, flavonoids and structurally diverse natural products and synthetic analogs [4,150].

### 7.1. ROS Pathway

ROS inducers are among the most well-characterized compounds that decrease levels of Sp TFs in cancer cells, and this response contributes to their overall anticancer activities. ROS inducers include phenethyl isothiocyanate (PEITC), benzyl isothiocyanate (BITC), celastrol, curcumin, betulinic acid, piperlongumine, penfluridol, the nitro aspirin GT-094, histone deacetylase (HDAC) inhibitors, hydrogen peroxide, ascorbic acid, arsenic trioxide, and t-butyl hydroperoxide [151,152,153,154,155,156,157,158,159,160,161,162,163,164,165,166]. In addition, several other compounds that target Sp1 downregulation including phloretin, honokiol, triptolide baicalin, quercetin, licochalcone, 7,8-dihydroxyflavone [167,168,169,170,171,172,173,174,175,176,177,178,179,180] also induce ROS [181,182,183,184,185,186] and may act in some cell lines through the ROS-Sp (downregulation) pathway. The mechanism associated with ROS-dependent downregulation of Sp TFs was determined over several years and was in part dependent on two separate and independent studies. Firstly, O’Hagan and coworkers reported that ROS induced genome-wide chromatin shifts of complexes containing CpG islands and this resulted in the downregulation of c-Myc [187]. A second study reported that treatment of breast cancer cells with an HDAC inhibitor induced expression of an Sp repressor gene ZBTB10 and this was accompanied by downregulation of miR-27b, which is part of the miRNA-23a-27a-24-2 cluster [188]. These results, coupled with extensive studies on various ROS inducers, conclude that the overall mechanism of Sp downregulation is linked to ROS-dependent downregulation of Myc and Myc-dependent miRNAs (including miR-27a) (Figure 5). This results in the induction of ZBTB10, which in turn competitively binds GC-rich promoters and displaces Sp TFs. ZBTB10 and related ZBTB genes do not have a transactivation domain, and this results in gene silencing at GC-rich sites at the expense of Sp TFs. Subsequent studies show that miR-27a also regulates expression of ZBTB34 [151,152,162,165,166]. This also results in decreased expression of Sp1, Sp3 and Sp4, which are self-regulated genes. In addition, cMyc also regulates miRNA-20a and miRNA-17-5p expression, which are part of the miRNA-17-92 cluster, and this results in induction of ZBTB4, which also represses expression of Sp TFs [189]. The results of these studies were confirmed by both overexpression and rescue experiments and also demonstrate that knockdown of cMyc also decreases levels of Sp1, Sp3 and Sp4.

### 7.2. Kinase/Phosphatase Pathway

Interestingly, the role of ZBTB-induced suppression was also observed downstream from drug-induced activation of kinases through the cannabinoid receptor. The synthetic CB receptor ligand WIN55,212-activates protein phosphatase 2A, resulting in miRNA-27a downregulation and activation of ZBTB10 in colon cancer cells [190] (Figure 5). Moreover, it was reported in that in breast cancer cells, betulinic acid also targeted the miRNA-279-ZBTB10 pathway through betulinic acid acting as a cannabinoid receptor ligand [191]. The antidiabetic drug metformin also induced Sp downregulation and like many other agents noted above, the mechanism was cell-context-dependent. In Panc1 cells, metformin-dependent downregulation of Sp TFs was due to mitogen-activated protein kinase phosphatase 1 (MKP-1) and MKP-5, which targeted miR-27a-ZBTB10, whereas in Panc28 and L3.6pL cells, metformin induced proteasome-dependent degradation of Sp1, Sp3 and Sp4 [192,193] (Figure 5). In addition, several other studies found that phosphatases induced Sp1 downregulation. For example, progesterone activation of progesterone receptor induced MKP1 and Sp1 downregulation [193] and both α-tocopherol succinate and hydrogen peroxide activated a phosphatase-JNK1 pathway that also decreased expression of Sp1 [194,195].

### 7.3. Proteasome-Dependent Degradation

Several studies have reported proteasome-dependent degradation of Sp1, Sp3 and Sp4 by a number of anticancer agents, including tolfenamic acid and related NSAIDs, betulinic acid and celecoxib [196,197,198,199,200,201]. The mechanisms of drug-induced degradation of Sp TFs by proteasomes has been investigated in other studies, which suggest that multiple pathways are involved. Sumoylated Sp1 can recruit the E3 ubiquitin ligase RING Finger protein 4 (RNF4) which undergo proteasome degradation; a similar pathway has also been observed for betulinic acid, which induces degradation of Sp1 and Sp3 [202,203,204,205]. Further studies on the role of sumoylation and other cofactors on degradation of Sp1, Sp3 and Sp4 need to be further investigated. Activation of caspases also plays a role in Sp degradation, and this pathway has been observed for several drugs including aspirin, retinoids, tolfenamic acid, and bortezomib; these effects are also cell-context-dependent [206,207,208,209,210,211,212]. Many of these studies have focused on the mechanisms associated with only one of the Sp proteins (usually Sp1), and the results clearly demonstrate that several mechanisms are operative.

### 7.4. Activation of Caspases

For example, cleavage of Sp1 by a retinoid in liver cancer cells involves induction of caspase-3 and transglutaminase [210] and caspases-2 and 3 in leukemia cells [209]. A role for caspase-3 activation in Sp1 degradation has been observed in other studies [211,212], whereas bortezomib was found to decrease Sp1, Sp3 and Sp4 in leukemia cells, and this was dependent on caspase-8 [179]. The zinc chelator N,N,N′,N′-tetrakis (2-pyridylmethyl) ethylenediamine (TPEN) sequesters zinc, and this results in activation of caspases-3,8, and 9 and downregulation of Sp1 [207]. Similar results were observed in colon cancer cells treated with aspirin [206], which also induced degradation of Sp1, Sp3 and Sp4, and treatment with tolfenamic acid [208] gave results similar to that observed for aspirin. This was confirmed in studies showing that activation of caspases and Sp downregulation by TPEN, aspirin and tolfenamic acid was reversed in cancer cells after cotreatment with zinc sulfate. Figure 6 illustrates the structurally diverse agents that downregulate Sp transcription factors via mechanisms outlined in Figure 5. These studies indicate that drugs targeting the pro-oncogenic Sp1, Sp3 and Sp4 act through multiple pathways that are cell-context-dependent.

## 8. Summary and Conclusions

There is increasing evidence that Sp1, Sp3 and Sp4 play an important role in multiple cancers and their prognostic importance spans their functional pro-oncogenic activities alone and in combination with miRNAs and lncRNAs. Genomic studies on these transcription factors and genes/pathways regulated by Sp1, Sp3 and Sp4 demonstrate their role in the growth, survival and migration/invasion of cancer cells and tumors, and this is consistent with their designation as non-oncogene addiction genes. Moreover, this designation is supported by interaction of Sp TFs with ncRNAs where their role is associated with enhancing pro-oncogenic pathways. There is also extensive evidence that multiple compounds, including approved drugs that are used for other diseases, induce downregulation or degradation of Sp1, Sp3 and Sp4, which is accompanied by inhibition of cell/tumor growth and invasion and induction of apoptosis. Anticancer agents that target Sp TFs are not yet clinically used for cancer chemotherapy and the clinical applications of these agents, including repurposed drugs, need to be evaluated in combination therapies.

## Figures and Tables

**Figure 1 ijms-24-05164-f001:**
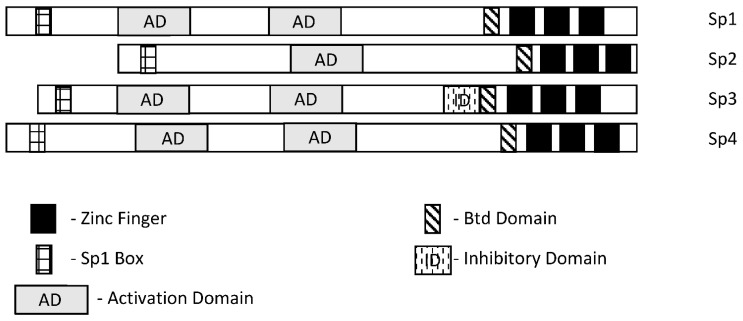
Schematic structures of Sp1, Sp2, Sp3 and Sp4 [1,2]. These transcription factors exhibit several common structural features; however, Sp3 expresses an inhibitory domain that results in gene-specific decreased expression in some cell lines.

**Figure 2 ijms-24-05164-f002:**
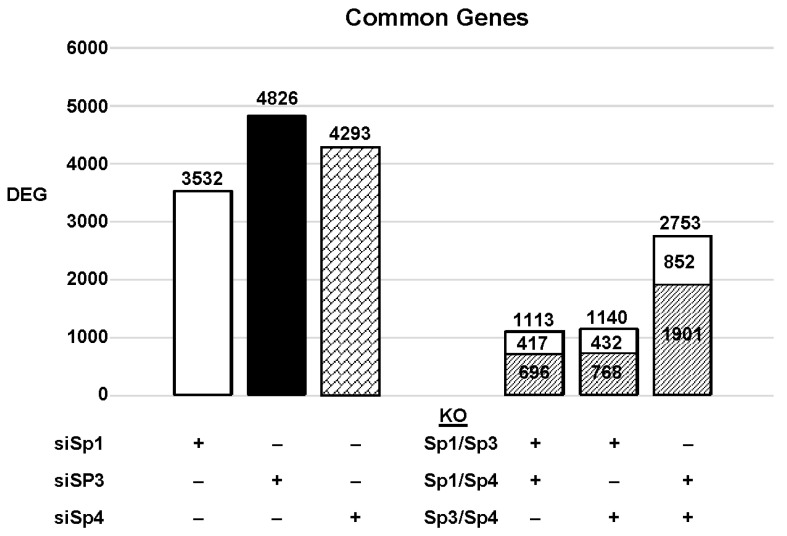
Sp knockdown and changes in gene expression [7]. Panc1 cells were transfected with siRNAs, and after Sp knockdown, the changes in gene expression and the genes commonly induced/repressed by siSp1/siSp3, siSp1/siSp4, and siSp4/siSp3 were determined (
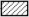
: decreased and 
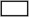
: increased expression in the double knockout groups).

**Figure 3 ijms-24-05164-f003:**
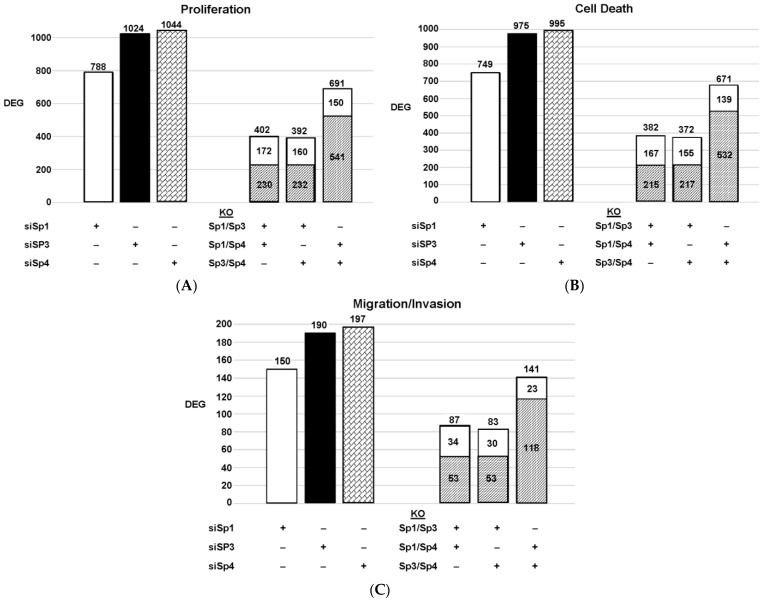
Effects of Sp knockdown by RNAi [7]: IPA analysis of the differentially expressed genes in Panc1 cells associated with cell proliferation (**A**), Annexin V staining (**B**), and invasion (**C**). In these same samples, the common genes observed after knockdown of Sp1/Sp3, Sp1/Sp4 and Sp3/Sp4 are given. (
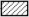
: decreased and 
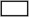
: increased expression in the double knockout group).

**Figure 4 ijms-24-05164-f004:**
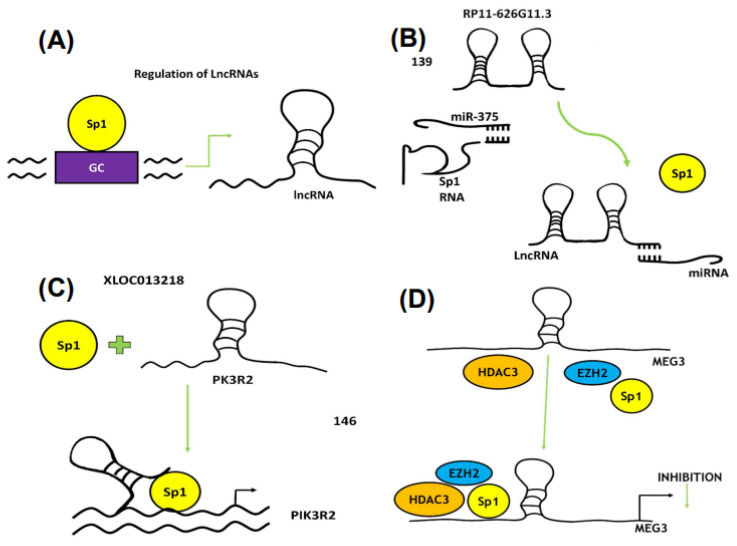
Interactions of Sp1 and lncRNAs. (**A**) Sp1 binds to the GC-rich promoter of a lncRNA to induce gene expression. (**B**) Silencing of Sp1 by miR-375 is reversed by competitive binding of a lncRNA to a miRNA. (**C**) Sp1 interacts with lncRNA XLOC013218 and forms an activation complex on the PIK3R2 gene promoter, whereas (**D**) MEG3 expression is inhibited by the EZH2/HDAC3/Sp1 complex.

**Figure 5 ijms-24-05164-f005:**
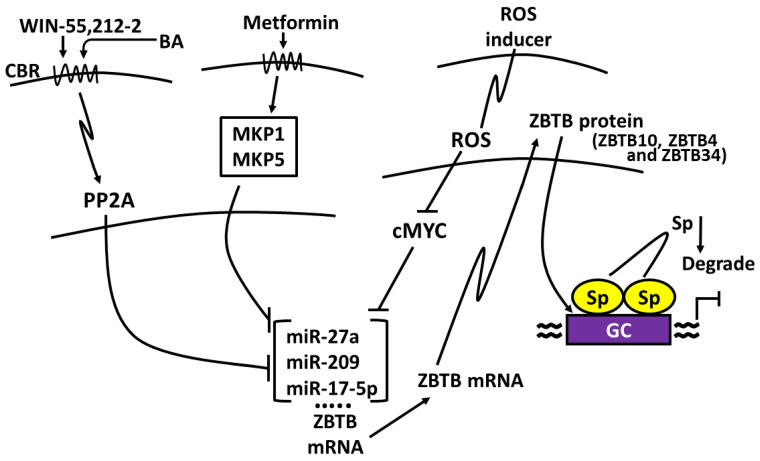
Mechanisms of Sp downregulation [149,150,160,162,164]: ROS-inducers target Myc [149,150] whereas Metformin and WIN target kinases [189,190,191] to activate ZBTB (via miRNA downregulation), which displace Sp TFs from GC-rich sites. ZBTB genes induced via these pathways include ZBTB10, ZBTB34 and ZBTB4.

**Figure 6 ijms-24-05164-f006:**
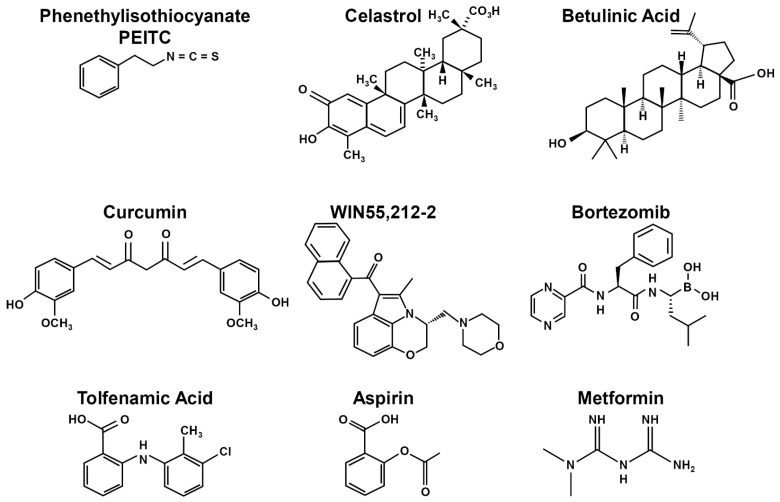
Structures of compounds that induce Sp downregulation. Compounds that induce downregulation of Sp1, Sp3 and Sp4 are structurally diverse and include PEITC [151], celastrol [153], betulinic acid [155,165,191], curcumin [154,158], WIN55,212-2 [190], bortezomib [179], tolfenamic acid [198,208], aspirin [206] and metformin [192,213].

**Table 1 ijms-24-05164-t001:** Clinical/prognostic Significance of Sp transcription factors.

Tumor	Sp TF	Prognosis	Refs.
Prostate	Sp1/Sp3/FLIP	Overexpression correlated with a high Gleason score and predicted recurrence	[14]
Esophageal squamous cell carcinoma	Sp1	High Sp1 predicts poor patient survival	[15]
Astrocytoma	Sp1	Poor patient prognosis	[16]
Bladder urothelial carcinoma	Sp	Poor clinical outcomes	[17]
Glioma	Sp1	Poor outcomes, higher expression in higher grades, immune invasion	[18,19,20]
Head and Neck	Sp3	Predicted poor survival	[21]
Pancreatic	Sp1 (Sp1/LOXL2)	Decreased survival, higher grade, dual prognostic factor (with LOXL2)	[22,23,24]
Oral squamous cell carcinoma	Sp1	Overexpressed and prometastatic	[25]
Gastric cancer	Sp1	Overexpressed, poor prognosis, increased in higher stages	[26,27,28,29,30]
Liver cancer	Sp1	Overexpressed, poor prognosis	[11,12]
Colin cancer	Sp1/Sp3	Overexpressed, decreased survival	[31,32]
Breast cancer	Sp1/Par3	Lower levels/advanced stage, poor prognosis	[33,34,35,36]
Lung cancer	Sp1	Variable prognosis, decreased Sp1 with increasing stage	[37,38,39]
Ovarian cancer	Sp1/DANCR	Sp1 overexpression in tumor, correlates with DANCR	[40]
Liver cancer	Sp2	Decreased survival	[13]

**Table 2 ijms-24-05164-t002:** MiRNA-Dependent inhibition of Sp. TFs.

miRNA	Sp TF	Tumor	Refs.
miRNA-29b	Sp1	Myeloid leukemia	[78,79]
miRNA-29b	Sp1	Multiple myeloma	[80]
miRNA-23b	Sp1	Multiple myeloma	[81]
miRNA-377	Sp1	Glioblastoma	[82]
miRNA-380-3p	Sp1	Neuroblastoma	[83]
miRNA-29b	Sp1	Tongue squamous cell carcinoma	[84]
miRNA-429	Sp1	Esophageal carcinoma	[85]
miRNA-506	Sp1/Sp3	Breast cancer cells	[86]
miRNA-27b	Sp1	Non-small cell lung cancer	[87]
miRNA-324-5p	Sp1	Hepatocellular carcinoma	[88]
miRNA-491-3p	Sp1	Hepatocellular carcinoma	[89]
miRNA-200b/200c	Sp1	Gastric cancer	[90]
miRNA-22	Sp1	Gastric cancer	[91]
miRNA-223	Sp1	Gastric cancer	[92]
miRNA-638	Sp1	Gastric cancer	[93]
miRNA-145/133a/133b	Sp1	Gastric cancer	[94]
miRNA-335	Sp1	Gastric cancer	[95]
miRNA-375	Sp1	Pancreatic adenocarcinoma	[96]
miRNA-375	Sp1	Colorectal cancer	[97]
miRNA-375-3p	Sp1	Colorectal cancer	[98]
miRNA-1224-5p	Sp1	Colorectal cancer	[99]
miRNA-382	Sp1	Colorectal cancer	[100]
miRNA-149	Sp1	Colorectal cancer	[32]
miRNA-429	Sp1	Renal cell adenocarcinoma	[101]
miRNA-137	Sp1	Bladder cancer	[102]
miRNA-375	Sp1	Squamous cervical cancer	[103]
miRNA-34a	Sp1	Hela cells	[104]
miRNA-330	Sp1	Prostate cancer	[105]

**Table 3 ijms-24-05164-t003:** Sp TFs regulate LncRNA expression in cancer cell.

Sp TF	LncRNA	Tumor	Ref.
Sp1	MIR155HG	Glioblastoma	[111]
Sp1	HOTTIP	Osteosarcoma	[112]
Sp1	Lnc00152	Retinoblastoma	[113]
Sp1	RNA TINCR	Gastric cancer	[114]
Sp1	LINC01638	Non-small cell lung cancer	[115]
Sp1	THAP7-AS1	Gastric cancer	[116]
Sp1	MELTF-AS1	Non-small cell lung cancer	[117]
Sp1	DUBR	Hepatocellular carcinoma	[53]
Sp1	HOTAIR	Hepatitis B virus	[118]
Sp1 a	PCAT6	Breast cancer	[119]
Sp1 a	PCAT19	Gastric cancer	[120]
Sp1 a	LINC00659	Gastric cancer	[121]
Sp1 a	LINC00520	Non-small cell lung cancer	[122]
Sp1 a	MIR155HG	Melanoma	[123]
Sp1a	CTBP1-AS2	Hepatocellular carcinoma	[124]
Sp1 a	HOXD-AS1	Cholangiocarcinoma	[125]
Sp1 a	LMCD-AS1	Osteosarcoma	[126]
Sp1 a	LINC00689	Osteosarcoma	[127]
Sp1 a	SNHG4	Prostate	[128]
Sp1/Sp3/Sp4	MALAT-1	Pancreatic cancer	[129]
Sp1 ab	MEG3	Pancreatic cancer	[130]
Sp1 b	SAMMSON	Thyroid carcinoma	[131]
Sp1 b	MALAT1	Lung adenocarcinoma	[132]
Sp1 b	HOTAIR	Hepatocellular carcinoma	[133]
Sp1 b	HOTAIR	NSCLC	[134]
Sp1	CRNDE	Hepatocellular carcinoma	[135]
Sp1 b	HOTAIR	Cutaneous squamous cell carcinoma	[55]
Sp1 ab	HOTAIRM1bc	Glioblastoma	[136]
Sp1	TUG1	Colorectal cancer	[137]
Sp1	POU3F3	Cervical cancer	[138]
Sp1 a	LINC00511	Glioma	[139]
Sp1 a	TINCR	Colorectal cancer	[140]
Sp1 a	RP11-626G113bc	Glioma	[141]
Sp1 a	MIR31HG	NSCLC	[142]
Sp1 a	SNHG22	Ovarian cancer	[143]

a: miR involved; c: +Line—Sp1; b: reciprocal.

## Data Availability

This is a review article summarizing published data.

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
