# Peer review of "Specificity Proteins (Sp) and Cancer"

_ijms, 2023, doi:10.3390/ijms24065164_

Round 1

Reviewer 1 Report

The current manuscript entitled: Specificity proteins (Sp) and cancer by Stephen Safe, summarized the role of Sp transcription factor as prognostic factor, involving cancer cell growth, metastasis, gene regulation and possible drug target. The author has done a great job in writing and including the most advanced research article related to Sp in this review. I have few suggestions to improve the MS.

1.The author should include few sentences in the abstract section about what the author wants to convey this review. For ex. In this Review, we discuss the…….  

2.In section 3. The author should describe the genes that are directly regulated by Sp in sub section. For ex, 3.1, 3.2 etc.

3. The author should elaborate the individual miRNA and LncRNA into more details with sub section.

4. Similarly in the section 6, include the drugs and describe the mechanism of action separately in sub section.

5. It is clearly known that Sp TF regulate gene expression by binding to GC rich region in the DNA. It would be good if the author can describe little more about enhancer driven gene expression by Sp.

6.The authors can improve these figures by redraw it. (Looks directly taken from the ref paper)

Author Response

Thank you for taking the time to review our manuscript, below please find our responses to your comments/suggestions:

1. The author should include few sentences in the abstract section about what the author wants to convey this review. For ex. In this Review, we discuss the……. 

Author Response: A few sentences have now been added.

2. In section 3. The author should describe the genes that are directly regulated by Sp in sub section. For ex, 3.1, 3.2 etc.

Author Response: We have identified some key Sp regulated genes; however, this was limited since there are several thousand Sp regulated genes as reported in (7).

3. The author should elaborate the individual miRNA and LncRNA into more details with sub section.

Author Response: We have summarized Sp-non-coding RNA interactions in two Tables. A discussion of each ncRNA would not add much to the review however, we have expanded our comments on the large number of different ncRNAs that cooperatively work with Sp transcription factors.

4. Similarly in the section 6, include the drugs and describe the mechanism of action separately in sub section.

Author Response: The different mechanisms outlined in the text have been separated and a new Figure 6 illustrates some of the drugs that target Sp downregulation.

5. It is clearly known that Sp TF regulate gene expression by binding to GC rich region in the DNA. It would be good if the author can describe little more about enhancer driven gene expression by Sp.

Author Response: This section has now been expanded in section 1.

6.The authors can improve these figures by redraw it. (Looks directly taken from the ref paper)

Author Response: The figures have now been improved and a new Figure 6 has been added.

Reviewer 2 Report

This review focus on discussing the relevant literatures about specificity protein (Sp) transcription factors Sp1, Sp3 and Sp4. Thus, the title is a bit misleading as six out of nine Sp genes are merely discussed here. There is no explanation of the structure and presentation order of the sections, making it very difficult for a non-Sp experts to understand this field and not suitable for the broad readership of the journal. Some specific issues I encountered are listed below. However, the key problem is the lack of a clear structure and relationship between each subsections.

One would expect the abstract to provide an overview of the manuscript, but the abstract only provides some background information about Sp1-4 (mainly Sp1/3/4) without the aim/objective/structure of the review. Consequently, it is difficult to get the purpose of this review from the abstract.

The introduction section at the beginning does not seem to provide the essential background information. It is better to use another section title. It would also be helpful if the author can provide an outline of the review in the section paragraph (Line 87-90).

Line 89: Should define "more recent studies" more specifically, e.g., last five years.

Table 2: Numbering of miRNA seems to be in a random order. Can they be presented in a more systematic manner?

I don't understand the abbreviation of DEG.

Author Response

Thank you for taking the time to review our manuscript, below please find our responses to your comments/suggestions:

One would expect the abstract to provide an overview of the manuscript, but the abstract only provides some background information about Sp1-4 (mainly Sp1/3/4) without the aim/objective/structure of the review. Consequently, it is difficult to get the purpose of this review from the abstract.

Author Response: The Abstract has now been improved (as also pointed out by Reviewer 1)

The introduction section at the beginning does not seem to provide the essential background information. It is better to use another section title. It would also be helpful if the author can provide an outline of the review in the section paragraph (Line 87-90).

Author Response: We have now rearranged this section and provided an outline of the Review within the text as indicated.

Line 89: Should define "more recent studies" more specifically, e.g., last five years.

Author Response: This change has now been made.

Table 2: Numbering of miRNA seems to be in a random order. Can they be presented in a more systematic manner?

Author Response: The number and designation of miRNAs and lncRNAs is random and confusing and not much can be done to improve this since nomenclature harmonization for ncRNAs has not been made.

I don't understand the abbreviation of DEG

Author Response: DEG = differentially expressed genes (now defined in the text)

Round 2

Reviewer 2 Report

Regarding Table 2, I meant the information are presented in a rather radom fashion. It would be helfpul if they are categorized by either miRNA numbers or tumor type or both. My suggestion is re-format this table according to tumor type, and show relevant miRNA in numeric order. 

Author Response

Thank you for looking at our revised paper, see our response to your comment below:

We could not revise Table 2 based on the numerical order of the miRNAs since the numbering system is not scientific (i.e., in order of discovery). The Table is now rearranged based on miRNA-Sp interactions with tumor type.